# The Role of Lignin in the Compartmentalization of Cadmium in Maize Roots Is Enhanced by Mycorrhiza

**DOI:** 10.3390/jof9080852

**Published:** 2023-08-15

**Authors:** Ruimin Lao, Yanying Guo, Weixia Hao, Wenjun Fang, Haiyan Li, Zhiwei Zhao, Tao Li

**Affiliations:** 1State Key Laboratory for Conservation and Utilization of Bio-Resources in Yunnan, Yunnan University, Kunming 650091, China; 2Kunming Dianchi & Plateau Lake Research Institute, Kunming 650228, China; 3Medical School of Kunming University of Science and Technology, Kunming 650504, China

**Keywords:** arbuscular mycorrhizal fungus (AMF), maize (*Zea mays*), cell walls (CWs), lignin, Cd tolerance

## Abstract

In nature, arbuscular mycorrhizal fungi (AMF) play a crucial role in the root systems of plants. They can help enhance the resistance of host plants by improving the compartmentalization of toxic metal contaminants in the cell walls (CWs). However, the functions and responses of various CW subfractions to mycorrhizal colonization under Cd exposure remain unknown. Here we conducted a study to investigate how Cd is stored in the cell walls of maize roots colonized by *Funneliformis mosseae*. Our findings indicate that inoculating the roots with AMF significantly lowers the amount of Cd in the maize shoots (63.6 ± 6.54 mg kg^−1^ vs. 45.3 ± 2.19 mg kg^−1^, *p* < 0.05) by retaining more Cd in the mycorrhized roots (224.0 ± 17.13 mg kg^−1^ vs. 289.5 ± 8.75 mg kg^−1^, *p* < 0.01). This reduces the adverse effects of excessive Cd on the maize plant. Additional research on the subcellular distribution of Cd showed that AMF colonization significantly improves the compartmentalization of 88.2% of Cd in the cell walls of maize roots, compared to the 80.8% of Cd associated with cell walls in the non-mycorrhizal controls. We observed that the presence of AMF did not increase the amount of Cd in pectin, a primary binding site for cell walls; however, it significantly enhanced the content of lignin and the proportion of Cd in the total root cell walls. This finding is consistent with the increased activity of lignin-related enzymes, such as PAL, 4CL, and laccase, which were also positively impacted by mycorrhizal colonization. Fourier transform infrared spectroscopy (FTIR) results revealed that AMF increased the number and types of functional groups, including −OH/−NH and carboxylate, which chelate Cd in the lignin. Our research shows that AMF can improve the ability of maize plants to tolerate Cd by reducing the amount of Cd transferred from the roots to the shoots. This is achieved by increasing the amount of lignin in the cell walls, which binds with Cd and prevents it from moving through the plant. This is accomplished by activating enzymes related to lignin synthesis and increasing the exposure of Cd-binding functional groups of lignin. However, more direct evidence on the immobilization of Cd in the mycorrhiza-altered cell wall subfractions is needed.

## 1. Introduction

Cadmium (Cd) is a naturally occurring element in the earth’s crust that causes serious harm to plants. High levels of Cd can adversely affect plant function by limiting photosynthesis [1], reducing nutrient absorption [2], and disrupting plant metabolism. As a result, excess Cd can ultimately lead to inhibited plant growth, lower crop yields, and decreased agricultural output, even without noticeable input use changes [3].

Plants possess several protective strategies to fight against Cd-related exposure. One of these is storing toxic Cd in cell walls or vacuoles to prevent the accumulation of heavy metals in intracellular organelles [4,5]. Plants also increase chelation to convert high concentrations of heavy metals to a less toxic state [6]. Additionally, plants can boost their antioxidant capacity to reduce damage caused by heavy metals [7] and activate anti-transporters to improve the removal of heavy metals [8,9].

One of the most crucial defense mechanisms in enhancing plant resistance to heavy metals is the compartmentalization of Cd in cell walls. This process prevents heavy metals from entering the cytoplasm. Previous studies have demonstrated that plant cells can tolerate heavy metals by modifying the structure and composition of their cell walls. These modifications involve thickening the cell walls [10,11], increasing the quantities of cell wall polysaccharides to absorb heavy metals [12], and exposing methylated pectin functional groups to bind more heavy metals [13,14]. Functional groups within the polysaccharide components of plant cell walls serve as binding and absorption sites for various heavy metals. One such example is *Ricinus communis*, which reduces the harmful effects of heavy metals by increasing the absorption of Cd and Zn by adding different functional groups such as –OH, amino, amide, and carboxyl groups to its cell walls [15]. Certain elements of cell walls, including cellulose, hemicellulose, and others, can also function as sites where heavy metals bind, particularly lignin and pectin [16,17,18,19].

Mycorrhizal fungi, an essential symbiont of plant roots, can also enhance the host plants’ ability to tolerate heavy metals. For example, both arbuscular mycorrhizal (*Funneliformis mosseae*) and ectomycorrhizal fungi (*Bovista limosa*) have been shown to significantly improve heavy metal tolerance within *Populus yunnanensis* [20,21]. Dark septate endophyte (DSE), a prevalent type of fungal colonizer, can also improve the ability of maize plants to tolerate cadmium (Cd). This is achieved by enhancing the compartmentation of Cd in cell walls and modifying its distribution at the subcellular level [22]. In addition, research has shown that arbuscular mycorrhizal fungus (AMF)—a type of symbiotic fungi that commonly colonize roots [23]—can improve how metals are stored in cell walls. This increases host plants’ tolerance to heavy metals [24]. However, it is unclear what specific roles and responses of various polysaccharides of cell walls have in AMF-enhanced cell wall compartmentalization to alleviate heavy metal exposure.

*Zea mays* was selected as a model plant for this study due to its widespread use in agriculture and frequent exposure to Cd. Additionally, *Zea mays* can adjust to environments with soil contaminated with cadmium (Cd). This feature makes it a valuable option for managing low- and medium-grade Cd-contaminated soils, whether for grain yield or energy production purposes [25,26]. In the present study, we aim to model the mycorrhizal symbiotic relationship between the host *Zea mays* and *F. mosseae* to explore (i) the functional roles of cell wall polysaccharides of host plants augmented by AMF in the enhancing compartmentalization of Cd and (ii) the physiological responses of host cell walls to AMF inoculation under Cd exposure.

## 2. Materials and Methods

### 2.1. Experimental Design

*Zea mays* Huidan4# seeds, commonly grown in Yunnan Province in China, were selected and purchased from Taifeng Seed Industry Co., Ltd. in Mangshi City, China. The seeds were sterilized with 75% ethanol and 10% NaClO for 5 min, rinsed three times with sterile water, and then transferred to sterile vermiculite for germination for 7 days. Uniform-size seedlings were selected and randomly divided into two groups. The mycorrhizal inoculation group (M+) were transplanted to pots containing 1150 g sterile vermiculite mixed with 250 g of fungal inoculant (*F. mosseae*, BGC YN05, 1511C0001 BGCAM0013, Institute of Plant Nutrition and Resources, Beijing Academy of Agriculture and Forestry, China) [27]. The non-mycorrhizal control group (M−) were inoculated with an equal size of inactivated AMF (121 °C, sterilized 3× for 2 h at an interval of 24 h). In the present study, 25 mg kg^−1^ Cd exposure (2.22 × 10^−4^ mol kg^−1^ Cd^2+^ by adding CdCl_2_·2.5H_2_O) (Cd25) to a medium-grade Cd-contaminated mine tailing soil was selected, in which mycorrhizal inoculation could significantly promote the growth of host maize [28]. Both the M+ and M− groups were then randomly divided into two sub-groups and transplanted into plastic pots containing 1400 g of culture substrate pretreated with either 0 (Cd0) or 25 mg kg^−1^ Cd^2+^ (Cd25). No CdCl_2_·2.5H_2_O was added to the Cd-free controls. There were a total of four treatments, which included maize that was either inoculated with (M+Cd0) or without (M−Cd0) AMF under Cd-free supplementation and maize inoculated with (M+Cd25) or without (M−Cd25) AMF under 25 mg kg^−1^ Cd^2+^ treatments. One maize seedling was cultured per pot. Ten replicates per treatment were conducted. The plants were grown in a greenhouse at 15–30 °C under natural light conditions for 43 days. All plants received 200 mL of 1/2 Hoagland’s solution every 7 days and were watered to maintain humidity at 70%.

### 2.2. Sample Collection

Prior to sample collection, plant height, basal diameter, and chlorophyll content (SPAD-502 Plus, Konica Minolta, Tokyo, Japan) were evaluated for each plant in all four treatments. Shoot and root samples of each maize plant were collected and weighed. A fresh subsample of all treated maize was collected, flash-frozen in liquid nitrogen, and stored at −80 °C for subsequent analyses. The remaining samples were dried at 60 °C for 72 h to yield a consistent weight and ground into powder using a mini-vegetation disintegrator (FZ102, Tianjin City Test Instrument Co., Ltd., Tianjin, China).

### 2.3. Determination of F. mosseae Colonization Intensity

Fresh root samples were washed in running tap water and cleaned in a 10% KOH (*w*/*v*) solution in a 90 °C water bath for 15 min. Translucent roots were then stained with 0.5% acid fuchsin [29]. The root colonization intensity of *F. mosseae* was assayed and scanned using the gridline intersect method under a light microscope (Olympus-CX31, Olympus Corporation, Tokyo, Japan). More than 300 intersects per sample were evaluated. We recorded the presence or absence of mycorrhizal structures at each point where the roots intersected a gridline. We assessed the level of colonization of AMF by calculating the ratio of the number of AMF-colonization intersects to the total intersects. To determine the overall mycorrhizal colonization intensity, we used the ratio of total mycorrhizal intersects (i.e., the total root intersects minus non-mycorrhizal intersects) to the total root intersects [30].

### 2.4. Root Cell Wall Extraction and Fractionation

Cell walls were extracted according to the method described by Shen et al. [22]. One gram of fresh root sample was ground in liquid nitrogen, mixed with ice-cold 75% ethanol, and incubated for 20 min at 4 °C. Samples were then centrifuged at 5000× *g* for 20 min. The precipitates were collected and sequentially washed with ice-cold acetone, methanol chloroform (*v*/*v* = 1), and ice-cold methanol. The precipitates were collected and assigned as cell wall (CW) fractions. Supernatants were pooled and deemed to be non-cell wall components (nCW). The above extracted CW and nCW fractions were freeze-dried and stored at 4 °C for further analyses. A ten mg quantity of the above freeze-dried CW was treated twice in a boiling water bath for 1 h, using 0.5% ammonium oxalate buffer (containing 0.1% NaBH_4_, pH = 4) and centrifuged at 10,000× *g* for 5 min. Supernatants from this step were pooled and marked as pectin fractions. The pellets were extracted three times with 4% KOH (containing 0.1% NaBH_4_) at room temperature for 24 h. After extraction, the supernatants were combined and labeled as hemicellulose 1 (HC1). The remaining pellets were then extracted three times using 24% KOH (which contained 0.1% NaBH_4_). The resulting final supernatant was labeled as hemicellulose 2 (HC2), while the precipitant was identified as a lignin component.

### 2.5. Measurement of Pectin and Lignin Content

The pectin content was determined according to the method described by Jia et al. [16]. A standard curve was created to determine pectin concentration by evaluating the OD values at 525 nm of serial dilutions of D (+) galactose uronic acid. One milliliter of the sample pectin was chilled in an ice bath for 10 min, mixed with a sulfuric acid–boric acid solution and a 0.15% resorcinol solution, and then measured for their OD values at 525 nm. The pectin concentrations were determined by comparing their OD values to the standard curve.

The content of lignin was determined according to the description by Acker et al. [31]. A 0.2 g quantity of CWs was mixed with 25 mL of 25% bromoacetyl acetate solution and 0.1 mL of 70% perchloric acid solution and incubated in a water bath at 70 °C for 30 min. To each sample, 2 M NaOH solution and 12 mL glacial acetic acid were then added. Finally, all extractions were diluted to 50 mL using glacial acetic acid and centrifuged at 1000× *g* for 5 min. The OD values of the supernatants were measured at 280 nm.

### 2.6. Determination of Cd Concentration

The Cd accumulated in the CWs, the non-CW fractions (i.e., the pooled supernatants), and 4 CW polysaccharide components was evaluated. First, all samples were digested in 5 mL HNO_3_ and 3 mL H_2_O_2_ as described in the China National Standards (GB 5009.15-2014). An atomic absorption spectrophotometer determined Cd content following our previously described methods (FAAS ZA-3000; Hitachi, Ltd., Tokyo, Japan) [32]. To assess a plant’s ability to move metals from roots to shoots, a translocation factor (TF) was utilized. The TF was calculated by comparing the metal concentration in the shoots to that in the corresponding roots [33].

### 2.7. Characterization of Functional Groups of Root CWs and CW Subfractions

The characterization of functional groups in the root CWs and 4 CW polysaccharide components of all 4 maize treatments was assessed using Fourier transform infrared spectroscopy (FTIR). A 2 mg quantity of the CWs and each CW subfraction (CW1, CW2, CW3) was mixed with 200 mg of KBr and then pressed into thin slices. FTIR (NicoletiS10, USA) was used to identify the functional groups involved in metal ion binding in CWs and 4 CW polysaccharides within the 4000–400 cm^−1^ range. The characteristic absorption peak of C−H at 2924 cm^−1^ (A_2924_) was used as a spectral reference. The ratio of the targeted absorption peaks to A_2924_ (marked as A/A_2924(treatments)_) was used to semi-quantitatively assess the changes of Cd binding via various functional groups of cell walls and 4 CW polysaccharide components (pectin, HC1, HC2, and lignin) [34]. Further, the mycorrhiza-responsive absorption peaks (*X*) of a given functional group in one CW subfraction were calculated. X was defined as the mycorrhizal groups minus the corresponding un-inoculating treatments under 25 mg kg^−1^ Cd supplementation based on the following formula:*X* = *A*/*A*_2924(M+Cd25)_ − *A*/*A*_2924(M−Cd25)_
where *A* refers to the absorption peak of a given functional group in one CW subfraction, and *A*_2924_ is the absorption peak of C−H at 2924 cm^−1^ in the same CW subfraction.

The peak shift of different functional groups in cell walls, pectin, and lignin components was quantitatively determined as the functional groups involved with Cd binding.

### 2.8. Assaying the Activity of Pectin- and Lignin-Related Enzymes

Five critical enzymes related to lignin synthesis (i.e., Phenylalanine ammonialyase, PAL; 4-Coumarate coenzyme Aligase, 4CL; and laccase) and pectin (Pectin methylase, PME; α1,4 galacto syltransferase, α1,4 GalT) metabolism were assessed at 4 °C according to the manufacturer’s protocol. Briefly, 0.1 g of fresh root tissues was ground into powder using liquid nitrogen and used to measure the activity of different enzymes, including PAL (Suzhou Keming Biotechnology Co., Ltd., Suzhou, China), 4CL (Beijing Solarbio Science &Technology Co., Ltd., Beijing, China), laccase activity (Suzhou Keming Biotechnology Co., Ltd., Suzhou, China), PME (Jiangsu Enzyme Immunoassay Industry Co., Ltd., Yancheng, China), and α1,4 GalT (Jiangsu Enzyme Immunoassay Industry Co., Ltd., Yancheng, China).

### 2.9. Statistical Analysis

All statistical analyses were performed using IBM Statistics 24.0 (SPSS, Inc., Chicago, IL, USA). First, the normality and homoscedasticity of the data were analyzed. We then used an independent sample *t*-test to analyze the significant differences in biomass, lignin content, and AMF colonization under Cd0 and Cd25 conditions. Two-way ANOVA was used to compare the interactive effects caused by the two variables: mycorrhizal inoculation and Cd treatment. All data were expressed as means ± standard errors (SE). All images were drawn with Origin 2018 (Originlab, Northampton, MA, USA).

## 3. Results

### 3.1. Colonization of F. mosseae in Maize Roots

All AMF-inoculated maize roots were successfully colonized by *F. mosseae* structures. Colonization of *F. mosseae* occurred mostly via the hyphae and arbuscules and sometimes via the hyphal coils, vesicles, and spores. AMF structures were not observed in the roots of the non-mycorrhizal controls. The addition of Cd did not noticeably change the colonization intensity of *F. mosseae* in maize roots (*p* > 0.05, *t*-test) (Table 1), with a total colonization intensity of 73.214 ± 0.024% and 77.615 ± 0.042% under 0 and 25 mg kg^−1^ Cd supplementation, respectively (Appendix A and Table 1).

### 3.2. Maize Growth Promoted by F. mosseae under Cd Exposure

In comparison to the M−Cd0 treatment, the addition of Cd did not significantly hinder the growth of M−Cd25 maize (Figure 1). This suggests that the exposure to 25 mg kg^−1^ of Cd may not cause stress in maize Huidan4#. However, the colonization of *F. mosseae* significantly improved the growth of maize by increasing biomass (Figure 1C), plant height (Figure 1D), basal perimeter (Figure 1E), and chlorophyll content (Figure 1F). This improvement was observed regardless of the addition of Cd and was higher than that of their respective non-mycorrhizal controls. For example, under the Cd25 treatments, the shoot and root biomass of M+Cd25 maize increased by 122.15% and 40.79%, respectively (Figure 1C).

### 3.3. The Enhanced Compartmentation of Cd in Cell Walls of F. mosseae Maize

When compared to the non-mycorrhizal controls, the M+Cd25 maize showed a significant reduction in Cd accumulation in the maize shoots under 25 mg kg^−1^ Cd exposure (45.34 ± 2.19 mg kg^−1^ Cd vs. 63.59 ± 6.54 mg kg^−1^ Cd, DW, *p* < 0.05, *t*-test) (Figure 2A). However, the mycorrhizal maize roots showed increased Cd accumulation (224.03 ± 17.13 mg kg^−1^ Cd vs. 289.48 ± 8.75 mg kg^−1^ Cd, DW, *p* < 0.01, *t*-test) (Figure 2A). AMF colonization also improved the Cd binding capacity of cell walls and reduced the translocation factor (TF) by 52% in the AMF-inoculated treatment compared to the M−Cd25 treatment (Figure 2B).

Data on the subcellular distribution of root Cd showed that there was a significantly higher proportion of Cd in the cell walls of the M+Cd25 group (88.2 ± 0.03%) than in the M−Cd25 group (80.8 ± 0.01%) (*p* < 0.05, *t*-test) (Figure 2D). Conversely, the accumulation of Cd in non-cell wall components (nCW) (including diverse cellular organisms) in mycorrhizal maize groups was significantly reduced (5.219 ± 0.73 mg kg^−1^ in M−Cd25 vs. 2.512 ± 0.24 mg kg^−1^, FW, in M+Cd25, *p* < 0.01, *t*-test) (Figure 2C).

### 3.4. Cd Accumulation in Cell Wall Polysaccharides

In maize roots, the majority of Cd was bound to the pectin fractions, which accounted for over 65% of Cd, irrespective of *F. mosseae* inoculation (as shown in Figure 3A,B). Nevertheless, when compared to M−Cd25, there was a significant decrease in the concentration of Cd in cell wall polysaccharides in M+Cd25. Pectin, in particular, was reduced by 26.4% (Figure 3A). The proportion of Cd accumulated in M+Cd25 pectin also significantly decreased, by 6.88% (71.2 ± 0.008% vs. 66.3 ± 0.006%, *p* < 0.01, *t*-test) (Figure 3B). In addition, the proportion of Cd in lignin to the total CWs significantly increased in lignin fractions from mycorrhizal roots (13.2 ± 0.002% vs. 16.7 ± 0.012%, *p* < 0.01, *t*-test) (Figure 3B). This increase resulted in a 0.34-time increase in the lignin content of M+Cd25 (5.8 ± 0.39 μg kg^−1^) compared to M−Cd25 (4.4 ± 0.27 μg kg^−1^) (*p* < 0.01, *t*-test) (Figure 3C). However, there was no significant change in the content of cell wall pectin observed in the two Cd-treated groups (Figure 3D), though mycorrhizal inoculation significantly increased the pectin content of M+Cd0 maize by 53.4% compared to M−Cd0.

### 3.5. Characteristics of Functional Groups of Cell Walls for Cd Binding

Compared to M−Cd25, the inoculation of *F. mosseae* (M+Cd25) induced changes in the functional groups of the cell walls, including lignins and pectins (Figure 4), indicating that these functional groups are responsive to mycorrhiza and may be play a role in metal binding. Notably, the most mycorrhiza-responsive carboxylates and amide I changes were observed in the cell walls, lignins, and pectins (Figure 4). Observations also revealed shifts in other functional groups, such as −CH and phosphates in the cell walls (Figure 4A); −OH/−NH in lignin components (Figure 4B); and −OH/−NH, amide I and sugar chain C−H groups in pectins (Figure 4C). In addition, we also observed that in the Cd-treated comparisons between M+Cd25 and M−Cd25, more types of functional groups shifted compared to the Cd-free comparisons between M+Cd0 and M−Cd0. For instance, the shifts of −CH and amide I in cell walls (Figure 4A) and −OH/−NH groups in lignins (Figure 4B) were observed in the comparisons of M+Cd25 and M−Cd25, but not in the comparisons of M+Cd0 and M−Cd0. In contrast to the above shifted characteristics of functional groups in cell walls and lignins, the types and numbers of the shifted functional groups significantly decreased in the pectin subfractions of M+Cd25 compared to M−Cd25. For example, the evident shifts of mycorrhiza-responsive −COOH and phosphates happened in the pectin subfractions in the two Cd-free comparisons between M+Cd0 and M−Cd0, but did not occur in the two Cd comparisons (Figure 4C). These results align with a significant decrease in Cd concentration in pectins in *F. mosseae*-colonized maize.

Furthermore, the FTIR results revealed that *F. mosseae* colonization (i.e., M+Cd25) led to changes in the absorption peaks of functional groups, potentially involved in Cd binding in the cell walls and CW polysaccharides when compared to the M−Cd25 treatments. Notably, there were increased absorption peaks of −COOR, acylamino, and carboxylate in cell walls, as well as −OH/−NH, acylamino and carboxylate in pectins and lignins (Appendix A).

### 3.6. Enzyme Activity Related to CW Polysaccharide Biosynthesis

Regardless of the colonization of *F. mosseae*, Cd addition increased the activity of enzymes that are involved in the production of lignin and pectin. Some of these enzymes included 4CL, laccase, PME, and α1,4 GalT. For example, compared to the Cd0 treatments, Cd supplementation significantly boosted laccase activity, a key enzyme for lignin synthesis. The increase was 5.92 times at M−Cd25 and 8.30 times at M+Cd25 in maize (Figure 5C). In M–Cd25 maize, the levels of two pectin biosynthetic enzymes, namely pectin methylesterase and α1,4-galacto syltransferase, increased by 21.81% and 22.47%, respectively, compared to M–Cd0 (Figure 5D,E). We observed that AMF affected lignin- and pectin-biosynthetic enzymes differently. When colonized by *F. mosseae*, the activity of lignin-biosynthetic enzymes such as phenylalanine ammonialyase and laccase was found to increase compared to M−Cd25 (Figure 5A,C). However, the activity of pectin-biosynthetic enzymes, namely PME and α1,4 GalT, significantly decreased by 8.1% and 9.4% (Figure 5D,E), which was consistent with the changes to their content of lignin and pectin. We also noted that the AMF-increased activity of 4CL and laccase may be dependent on Cd exposure (Figure 5B,C).

## 4. Discussion

Recently, many studies have shown that AMF colonization can increase the absorption area of roots, promote growth, and enhance heavy metal tolerance of host plants [35,36]. Understanding how cell wall components enhance heavy metal tolerance in plants is crucial, as these components serve as the initial line of defense for plants [6,24]. In our study, we exposed maize inoculated with AMF to 25 mg kg^−1^ of Cd, which resulted in increased growth. These improvements in growth seem to be attributed to a reduction in Cd concentration in the plant shoots, likely due to AMF limiting the translocation of Cd from roots to shoots. Consequently, more Cd accumulated in the maize roots, leading to decreased damages caused by Cd exposure in the plant shoots. These findings are consistent with previous conclusions that AMF colonization not only enhances the growth of host plants but also significantly improves their tolerance to heavy metals [37]. It is worth noting that these mycorrhizal effects depend on both the genetic characteristics of the host plants and the colonization of AMF [33]. Additional research has confirmed that mycorrhizal colonization can regulate the accumulation and compartmentalization of heavy metals in cell walls, contributing to the plant’s tolerance under heavy metal exposure. For example, AMF enhances the Pb tolerance of *Medicago truncatula* by remodeling cell wall components and enhancing antioxidant capacity as compared to uninoculated controls [38]. Similarly, Jiang et al. found that AMF-colonized *Medicago truncatula* showed increased compartmentalization and detoxification of Cd ions in the root cell walls [39]. Our results also found that AMF colonization significantly increased the accumulated concentration of Cd in the cell walls of maize roots, while reducing the Cd concentration in the non-cell wall components. These results suggest that the AMF-enhanced compartmentalization of Cd in the host plant’s cell walls is a crucial strategy for enhancing heavy metal tolerance in mycorrhizal host plants.

We observed that regardless of AMF colonization, up to 60% of Cd was accumulated in the pectin of maize cell walls. These findings align with previous research that identified pectin as the primary binding site for various heavy metals [40]. In a separate study, Hou et al. discovered that vanadium was predominantly stored in the pectin of cell walls in maize roots, particularly in alkali-soluble pectin, which exhibited the highest vanadium binding capacity [19]. Similarly, Jia et al. found that Cd exposure led to the induction and promotion of pectin-related genes, SIQUA1 and SIPME1, as well as increased activity of pectin methylase [16]. Contrary to previous assumptions, our research suggests that under Cd exposure, the pectin fraction of cell walls may not necessarily be the primary site of responsiveness to AMF colonization. Interestingly, when exposed to 25 mg kg^−1^ of Cd supplementation, AMF colonization had no significant effects on pectin content but did significantly increase the lignin content in the cell walls of maize roots.

Furthermore, previous reports have indicated that mycorrhizal colonization leads to an increase in the proportion of Cd associated with lignin. Lignin, a common aromatic heteropolymer found in plants’ secondary cell walls, has been found to play a crucial role in Cd’s absorption, transport, and tolerance in various plant species [17]. For instance, *Verbena bonariensis* demonstrates a significant increase in lignin-related enzyme activity, such as chalcone synthase (CHS) and anthocyanidin synthase (ANS), under excessive Cd exposure, thereby promoting lignin metabolism in cell walls [12]. Similarly, the application of salicylic acid has been shown to enhance lignin biosynthesis, promote cell wall synthesis, and prevent Cd entry through the root cells of rice (*Oryza sativa*) [41] and buckwheat (*Fagopyrum esculentum*) following Cd exposure [42]. In our study, we also observed that while inoculation alone or Cd exposure alone did not lead to significant changes in lignin content, the effect of AMF on lignin content depended on Cd exposure, which is in line with the findings reported by Gao et al. [24]. However, this contradicts the down-regulation of laccase encoding genes in mycorrhizal *Vaccinium myrtillus* exposed to Cd noted by Casarrubia et al. [43]. Laccases are multi-copper oxidases involved in the oxidative polymerization of lignin.

In regard to how AMF regulates lignin to enhance the Cd compartmentalization of maize cell wall, our findings demonstrate that *F. mosseae* significantly increases the activity of enzymes related to lignin synthesis in maize cell walls, including PAL, 4CL, and laccase, under Cd treatment. These results align with the typical response of various plants to different heavy metal exposures. By promoting the synthesis and thickening of cell walls, a significant amount of lignin can effectively prevent the entry of heavy metals into plant cells, ultimately improving the plant’s tolerance to heavy metals [44,45]. In 2018, Qiu and colleagues discovered that *Arabidopsis thaliana* can increase the expression of cinnamyl alcohol dehydrogenase following Cd exposure [46]. This enzyme is connected to the production of lignin, a vital component in the plant’s biosynthesis process. Cd can also induce significant lignification in the roots of *Medicago sativa*, leading to an enhanced accumulation of Cd^2+^ in the cell wall [47]. Similarly, the high expression of genes related to cell wall lignification, such as phenylalanine ammonia lyase, peroxidase, and laccase, is associated with enhancing plant Cd tolerance [48]. Therefore, the promotion of changes in lignin within the cell walls represents a common strategy for mycorrhizal plants to respond to heavy metal exposure.

The inoculation of AMF has a significant impact on the functional groups found in the cell walls’ polysaccharide components. These negatively charged functional groups are important for binding heavy metals and compartmentalizing them within the cell walls. During our research, we noticed clear shifts in the functional groups of M+Cd25, specifically –CH, amide I, and C−O−S/C−O/C−O−P groups in the cell walls and –OH/–NH in lignin. These groups act as binding sites for metals that are responsive to AMF. Many studies have shown that these negatively charged functional groups, such as hydroxyl and carboxyl groups, can bind to divalent metals, effectively immobilizing heavy metals [49]. For example, the functional groups N−H, C=O, C−N, and O−H in the cell wall of *Camellia sinensis* can provide binding sites for Cd^2+^ [50]. Similarly, under Cd exposure, the absorption peaks of −OH, C−H, and C−O in the cell wall of *Salix matsudana* have been shown to experience noticeable changes [34].

## 5. Conclusions

During exposure to Cd supplementation, *F. mosseae* successfully colonized maize roots and significantly improved compartmentalization, retaining 88.2% of Cd within the cell walls. This process effectively restricted the migration of Cd, which was absorbed by the roots, to the shoots, resulting in a substantial reduction in Cd concentration in the shoots. Consequently, the Cd tolerance of maize was greatly enhanced. While pectin remains the primary location of Cd binding, the presence of AMF caused a noticeable decrease in the amount and percentage of Cd accumulated in pectin. In contrast, there was a significant increase in the content of lignin components and their proportion of Cd. Moreover, the colonization of *F. mosseae* led to a significant rise in activity of lignin-related enzymes, such as PAL, 4CL, and laccase, which in turn promoted the development of more functional groups of lignin capable of binding Cd. This study has revealed that AMF enhances the ability of maize cell walls to compartmentalize and bind Cd by modifying the lignin within the cell walls rather than affecting pectin. However, to gain a deeper understanding of the mechanisms behind AMF-induced cell wall-related genes, further experiments are necessary. Additionally, more direct evidence regarding the immobilization of Cd in the mycorrhiza-altered cell wall subfractions is needed, gathered using inductively coupled plasma mass spectrometry (ICP-MS), scanning electron microscopy equipped with energy-dispersive spectroscopy (SEM−EDS), transmission electron microscopy equipped with energy-dispersive spectroscopy (TEM−EDS), and X-ray-absorption fine structure (XAFS) technologies [51].

## Figures and Tables

**Figure 1 jof-09-00852-f001:**
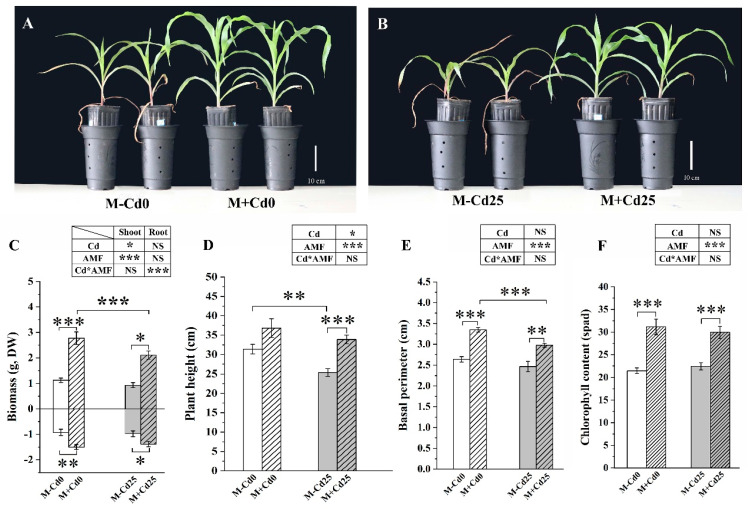
The growth (**A**,**B**), biomass (**C**), plant height (**D**), basal perimeter (**E**), and chlorophyll content (**F**) of maize plants evaluated based on two conditions: those inoculated with *F. mosseae* (M+) and those not inoculated (M−), and under two Cd supplementation levels: 0 mg kg^−1^ (Cd0) and 25 mg kg^−1^ (Cd25) for a period of 50 days. The data presented in the results represent the mean with standard error (*n* = 8), and statistically significant differences are indicated with asterisks (*). The level of significance is denoted as follows: (*, *p* < 0.05; **, *p* < 0.01; ***, *p* < 0.001, *t*-test). NS indicates no statistically significant difference (*p* > 0.05, two-way ANOVA).

**Figure 2 jof-09-00852-f002:**
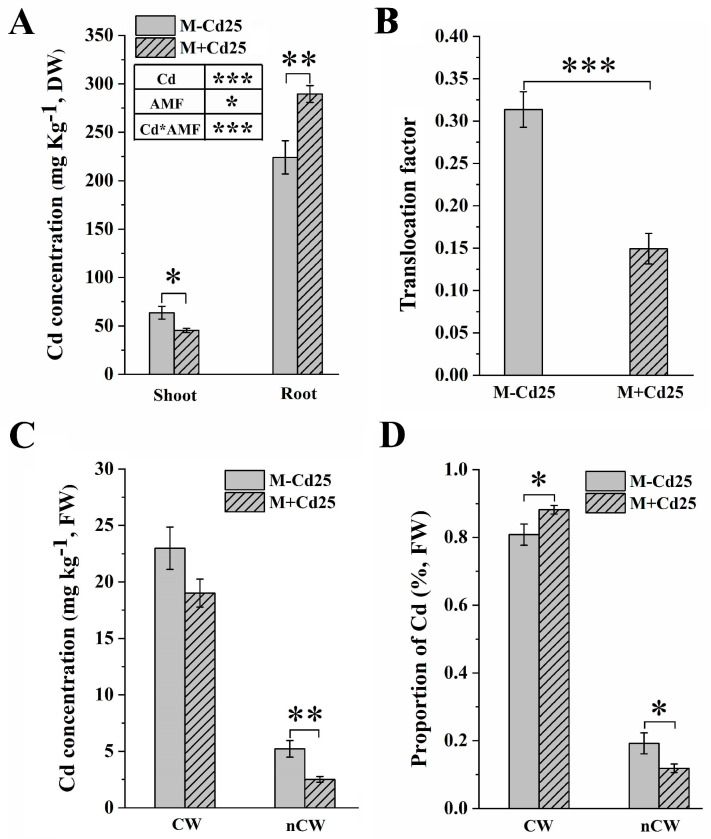
This data shows the concentration (**A**) and translocation factor of Cd (**B**) that accumulated in maize dry weight (DW), as well as the Cd concentration in maize fresh weight (FW) (**C**) and the Cd proportion (**D**) of various root subcellular fractions. The maize was either inoculated (M+) or not inoculated (M−) with *F. mosseae* and treated with Cd0 or Cd25. The data presented in the results represent the mean with standard error (*n* = 8), and statistically significant differences are indicated with asterisks (*). The level of significance is denoted as follows: (*, *p* < 0.05; **, *p* < 0.01; ***, *p* < 0.001, *t*-test). The subcellular fractions were divided into CW (cell walls) and nCW (remaining fractions excluding cell wall components).

**Figure 3 jof-09-00852-f003:**
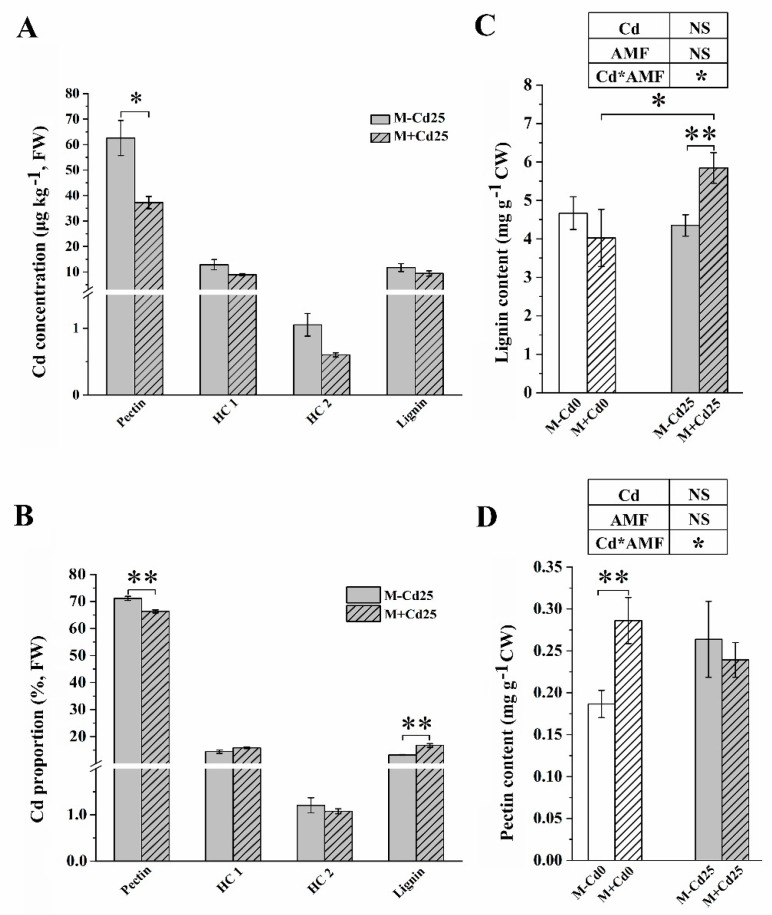
Cd concentration (**A**), Cd proportion (**B**), lignin content (**C**), and pectin content (**D**) in the cell wall (measured in fresh weight, FW) of roots that were either inoculated (M+) or not inoculated (M−) with *F. mosseae,* and exposed to either 0 mg kg^−1^ (Cd0) or 25 mg kg^−1^ Cd (Cd25) supplementation. The data presented are the means ± SE ((**A**,**B**), *n* = 4; (**C**,**D**), *n* = 8), and statistical significance is indicated by asterisks (* for *p* < 0.05, ** for *p* < 0.01, *t*-test). NS indicates no statistically significant difference (*p* > 0.05, two-way ANOVA).

**Figure 4 jof-09-00852-f004:**
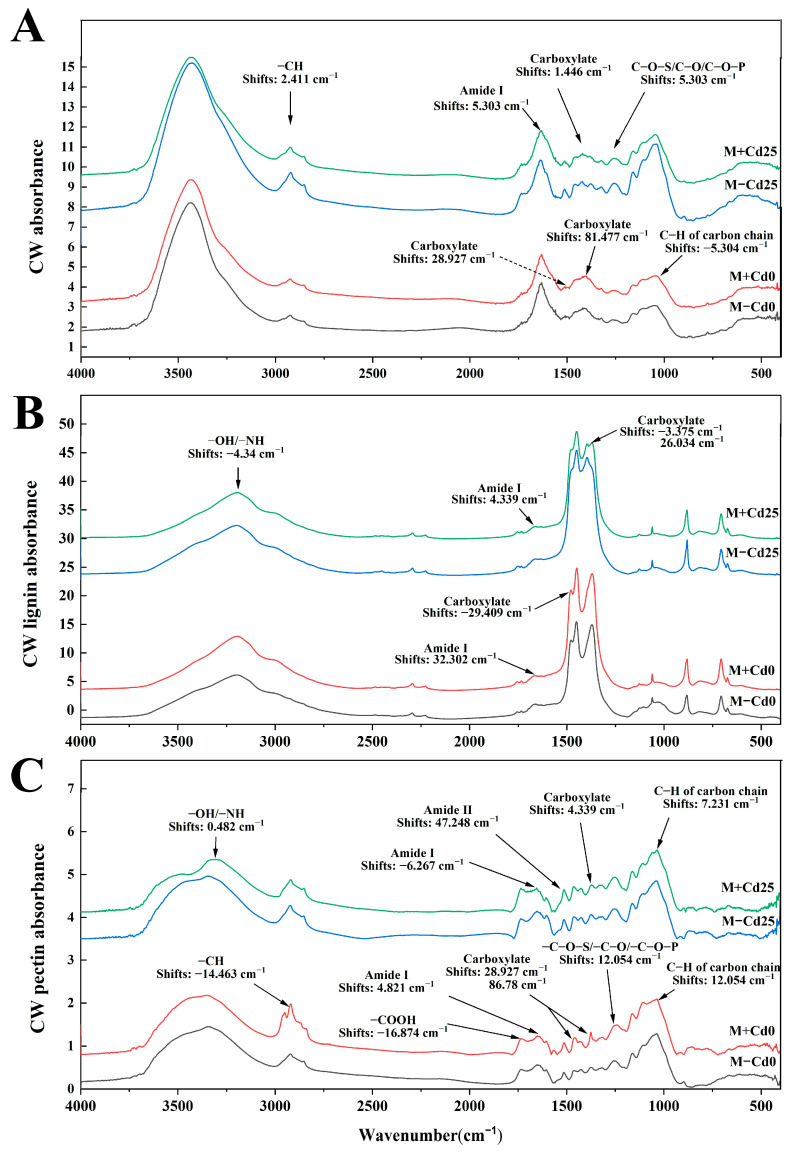
FTIR characteristics of functional groups with Cd binding potential from total cell wall (**A**), lignin (**B**) and pectin fractions (**C**) of un-inoculated (M−) or inoculated (M+) maize roots under 0 mg kg^−1^ and 25 mg kg^−1^ Cd supplementation.

**Figure 5 jof-09-00852-f005:**
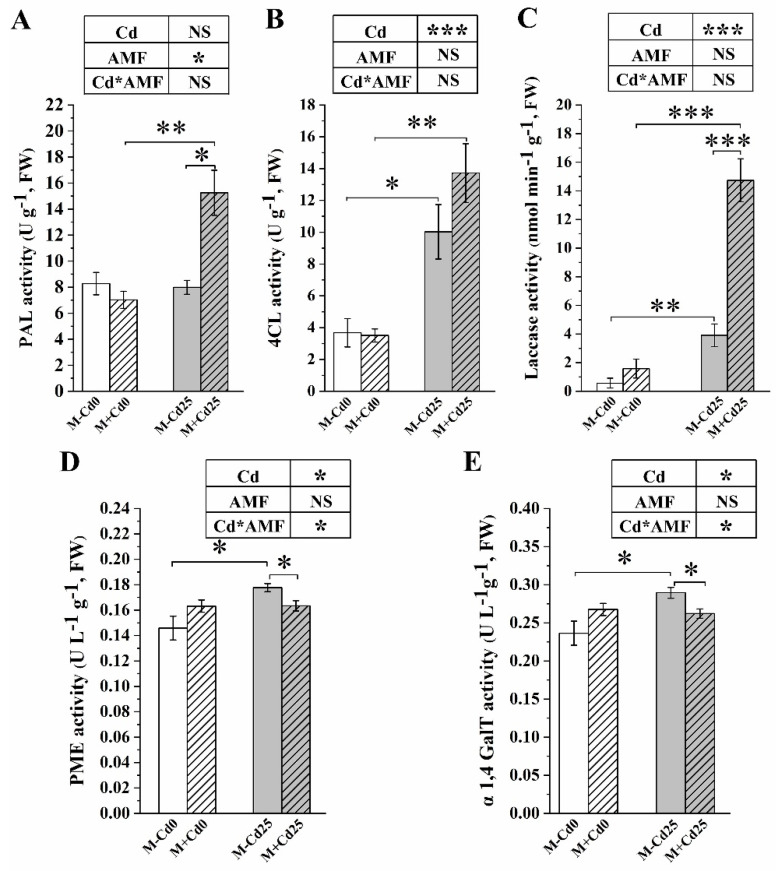
Activities of cell wall-related enzymes of maize roots inoculated (M+) or not inoculated (M−) with *F. mosseae* under 0 (Cd0) and 25 mg kg^−1^ Cd supplementation (Cd25). Phenylalanine ammonialyase (PAL) (**A**), 4-Coumarate:CoA ligase activity (4CL) (**B**), laccase activity (**C**), pectin methylase (PME) (**D**) and α1,4 galacto syltransferase (α1,4 GalT) (**E**). The results are presented as means ± SE (*n* = 4), with statistically significant differences indicated by * (*p* < 0.05), ** (*p* < 0.01) or *** (*p* < 0.001) using a *t*-test. NS indicates no statistically significant difference (*p* > 0.05, two-way ANOVA).

**Table 1 jof-09-00852-t001:** Colonization intensity of AMF in the maize roots inoculated with *F. mosseae* under Cd0 and Cd25 treatments. Data represent means ± SE (*n* = 4, *t*-test). Different lowercase letters in the same column indicate statistically significant difference.

Treatments	Colonization Intensity (%)				
	Hypha	Arbuscule	Vesicle	Hyphal Coil	Total
M+Cd0	53.70 ± 0.040 a	57.04 ± 0.038 a	0.39 ± 0.002 a	1.01 ± 0.003 a	73.21 ± 0.024 a
M+Cd25	42.32 ± 0.059 a	62.76 ± 0.039 a	0.12 ± 0.001 a	0.14 ± 0.001 a	77.61 ± 0.042 a

## Data Availability

All relevant data are within the manuscript.

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
