# Peer review of "The Role of Lignin in the Compartmentalization of Cadmium in Maize Roots Is Enhanced by Mycorrhiza"

_jof, 2023, doi:10.3390/jof9080852_

Round 1
Reviewer 1 Report
The manuscript needs writing improvement, many grammatical issues, wrong conjugation etc. However, the study appears to have been conducted thoroughly and with accuracy, and provide enough proof to link the sequestration of Cd in maize root by AM.
I would still be more careful in the conclusion as the results don't prove a direct link of Cd sequestration by AM or just a indirect consequence of increased lignin biosynthesis by AM therefore sequestering more Cd. Please be careful in the conclusion.
In general, I would not use the word stress. The plants seems to grow fine and don't have any sign of stress.
In https://www.tandfonline.com/doi/full/10.1080/15320383.2015.981648 the Cd IC50, is between 500 and 700mg/kg for wheat, Cabbage or radish.
In https://www.sciencedirect.com/science/article/pii/S0269749122002494?via%3Dihub#bib62
example of bio-remediation show absorption capacity of also hundreds of mg/kg
Please either show that the 25 mg/kg used in your study is a stress condition or if another publication prove this fact. Or remove the word stress from the manuscript.
This doesn't change the validity of your results.
Some results are still a bit confusing and need some improvement on the presentation.
Specific issues:
L219, wrong order of the words I presume, corrected sentence below:
reduced the accumulation of Cd by 25 mg kg−1 in the maize shoots under Cd stress (45.34 ± 2.19 mg kg−1)
Fig2: Please change the ABCD of the figure it is in the wrong order.
How do you calculate the translocation factor of Cd ? Please add in the text or M&M
Fig3: Please change the ABCD of the figure it is in the wrong order.
You show in Fig2 an increase of Cd in the CW but here you show a decrease of Cd in Pectin (5ug/kg/FW) and an increase in lignin (3.5ug/kh/FW).
Then how do you explain the general increase of Cd in mycorrhized roots ?
Part 3.5: I am very sorry, but this section is impossible to understand. please fully re-write the text. Go step by step and show what are the major differences.
Define Offset, what represent a positive vs negative offset.
L258-259 You cannot say that. Increased in the offset of functional groups within the cell walls, just show more functional group that's it.
How can you determine that any offset is due to Cd ? and not another component ?
If there are offsets in non Cd conditions then what you are looking at are just functional group. Doesn't mean they will bind Cd.
Please just re-explain the entire part.
in 2.7. Characterization of functional groups of root CWs and CW subfractions how is X used in the paragraph and the Fig 3 ?
L270-271 what decrease ? COOH ?
Part 3.6 is good
The manuscript needs writing improvement, many grammatical issues, wrong conjugation etc
Reviewer 2 Report
Review report ID jof-2504783_10072023:
Summary
Title: The role of lignin in the compartmentalization of cadmium in maize roots is enhanced by mycorrhiza
The paper describes how arbuscular mycorrhizal fungi (AMF) play a crucial role in the root systems of plants. The authors conducted a study to investigate how Cd is stored in the cell walls of maize roots colonized by Funneliformis mosseae. Their findings indicate that inoculating the roots with AMF lowers the amount of Cd in the maize shoots (63.6 ± 6.54 mg kg−1 vs. 45.3 ± 2.19 mg kg−1, p < 0.05) (28%) by retaining more Cd in the mycorrhized roots (224.0 ± 17.13 mg kg−1 vs. 289.5 ± 8.75 mg kg−1, p < 0.01) (29%). Additional research on the subcellular distribution of Cd showed that AMF colonization significantly improves the compartmentalization of 88.2% of Cd in the cell walls of maize roots, compared to the 80.8% of Cd associated with CWs in non-mycorrhizal controls. They claim that the presence of AMF did not increase the amount of Cd in pectin, a primary binding site for cell walls. However, it enhanced the content of lignin and the proportion of Cd in the total CWs-polysaccharides of the root cell walls. This finding was consistent with the increased activity of lignin-related enzymes, such as PAL, 4CL, and laccase which were also positively impacted by mycorrhizal colonization.
General concept comments
The Article writing could benefit from some improvements. There are many instances in which the described data don´t match with the figures. Figures captions not clear enough for the reader and in several instances don´t state properly what is displayed or omit important information. An English revision would be also needed to improve the clarity of the text.
For simplicity, I´d recommend to name the different experimental conditions as it follows:
M-Cd0 = Control C
M+Cd0 = M
M-Cd25 = Cd
M+Cd25 = M + Cd
Minor comments
Could the authors explain why the choose maize as model and why Cd at 25mg/kg-1?
Line 190. “All images were obtained through Origin 2018 (Originlab, Northampton, MA, USA)”. Could you explain the meaning of this sentence?
Line 199. The authors refer to supplementary figure (Figure S1. Typical structures of AM fungus in the roots of maize inoculated with F. mosseae under 0 (Cd0) and 25 mg kg−1 Cd (Cd25) stress for 50 days. These figures include arbuscules (A), hyphae (H), hyphal coils (HC), vesicles (V), and spores (S).) The figure does not explain clearly which structures, if any, are from Cd stressed roots. Could you please indicate that?
Line 212. Figure 1. Please edit captions. If experimental conditions are renamed, please edit accordingly.
Growth (A, B), biomass (C), plant height (D), basal perimeter (E), and chlorophyll content (F) of maize plants inoculated (M+) or non-inoculated (M−) with F. mosseae under 0 (Cd0) or 25 mg kg−1 Cd stress (Cd25) for 50 days. Data represent means ± SE (n = 8). * indicates statistically significant difference (*, p < 0.05; **, p < 0.01; ***, p < 0.001, t-test).
Line 227-231. Figure 2. The text of figure 2 is confusing and difficult to follow. It mentioned Cd0 treatment, but there is no data of this on the graphs, there is no mention of the meaning of DW and FW. I´d recommend editing to clarify it. There are some discrepancies between the text of figure 2 and the actual data displayed on graphs, for instance, figure 2b is not translocations factor. Please review, comment, and correct it when need it.
Lines 234-237. The authors claim that “the accumulation of Cd in non-cell wall components (nCW) (e.g., diverse cellular organisms) in mycorrhizal maize groups was significantly reduced (5.219 ± 0.73 mg kg−1 in M−Cd25 vs. 2.512 ± 0.24 236 mg kg−1 in M+Cd25, p < 0.01, t-test) (Fig 2C)”. The numbers don´t match with the graph. Could you explain?
Lines 239 to 250. I find this part of the text quite difficult to follow, confusing and the description does not match with the graphs, for instance, “The pectin fractions of maize roots were Cd's main binding sites, which accounted for more than 65% of Cd, regardless of F. mosseae colonization (Fig. 3A, B)”, where 3B is lignin content on the figure. Please correct and edit where need it.
Lines 245-247. The authors claim that “In lignin fractions of mycorrhizal 245 roots, the proportion of Cd was significantly increased (13.21 ± 0.002% vs. 16.72 ± 0.012%, 246 p < 0.01, t-test) (Fig. 3C).” Could you explain how this was calculated?
Lines 252-255. Figure 3. Again, the figure´s caption does not match the graph data, for instance Fig 3B is not Cd proportion. Please check n for each subfigure after editing. Please change “* indicate significant differences” to “* indicates statistically significant difference”. Please indicate the meaning of “FW”
Lines 260 to 262. The sentence lacks meaning.
Lines 273 to 278. I find very difficult to understand the meaning behind the text. Could you please elaborate?
Regarding Table S2, it is not clear to me what is reflecting, Cd content or content of functional groups with potential to bind Cd. Could you please elaborate as well?
Figure 4. For clarity I´d recommended editing. I´d suggest:
“FTIR characterization of functional groups with Cd binding potential from total cell wall (A), lignin (B) and pectin fractions (C) of non-inoculated (M-) or inoculated (M+) maize roots under 0 or 25 mg kg−1 Cd exposure”.
Lines 285-286. Please correct “biosyntheses”.
Line 296. Please correct 4-GalT (α-1,4 galacto syltransferase (α-1,4 GalT))
Line 303. “4. Discussion” should be in the next line.
Mayor comments
Table 1. The text is cut out in each cell of the table. Please correct it and explain the meaning of the text (a). The authors claimed that “the addition of Cd did not notably change the colonization intensity of F. mosseae in maize roots”, but looking at the table, we can see that Hypha structures were affected by Cd more than 10%, Arbuscules 9%, Vesicle structures were 3 times less common under Cd treatment and Hyphal coil 7 times. Could you explain this discrepancy on data and text? Could you explain how the total colonization was calculated?
Lines 207- 209. The authors claimed that “Compared to M-Cd0, F. mosseae colonization completely alleviated the inhibitory effects of stressed Cd on maize growth, with significantly increased shoot and root biomass by 83% and 53%, respectively (fig.1C). Taking a look to figure 1, we can see that there is no significant difference on biomass, basal perimeter or chlorophyl content between the control (M-Cd0) and Cd exposure (M-Cd 25). The only parameter that shows a significant difference is plant height. This suggests that the inoculation of F. mosseae is beneficial for the growth of the plant, as previously described, but is not alleviating any inhibitory effect from Cd. To see inhibitory effects and its compensation by the AMF inoculation I´d suggest using different Cd concentrations in a new set of experiments.
Line 222 -225. The authors claim that “AMF colonization enhanced the Cd binding capacity of cell walls, but figure 2B shows the opposite. The Cd concentration in CW from non-inoculated plants is bigger than those inoculated. Same happens in nCW fraction. Could you explain?
Figure 3B shows that, both, inoculation (M+ Cd0) or Cd exposure (M-Cd25), don´t increase lignin content independently, and it is the combination of Cd25 + inoculation what increases lignin content. This suggest that the AMF effect on lignin content is dependent on Cd exposure. This increase in lignin content due to M+Cd25 does not translate into a bigger Cd concertation in lignin fraction (Fig 3A). Could you explain that?
Lines 285-286. The authors claim that “Cd stress increased the activity of enzymes related to the biosyntheses of lignin and pectin”. This is not the case for PAL (Figure 4A), could you explain?
Lines 292- 297. The authors claim that “F. mosseae colonization increased the activity of lignin-biosynthetic enzymes, particularly phenylalnine ammonialyase, 4-coumarate:CoA ligase, and laccase, compared to M-Cd25 (Fig 5A, C). But looking at figure 5, we can see that inoculation itself (M+Cd0) does not induce change in these enzymes’ activities. Figure 5A shows that the combination of Cd and F. mosseae increases PAL activity. Fig 5B shows that Cd is the inducer of 4CL activity and Figure 5 C shows that Cd induces Lacasse activities which is even bigger with the combination of Cd and the AMF. Could you explain this discrepancy?
Additionally, the authors claim that “the activity of pectin-biosynthetic enzymes, i.e., PME, and 4-GalT, significantly decreased by 9.4% and 0.64%”, by the inoculation of the AMF. But Looking at Fig 5D and E, we can see that the inoculation of F. mosseae (M+Cd0) does not affect the enzymatic activities, that Cd (M- Cd25) is an inducer of both activities and the combination of Cd and inoculation does not affect the enzymatic activities when compare with the controls. Could you explain this discrepancy?
Lines 307 to 319. The authors claim that “Our study found that when exposed to 25 mg kg-1 of Cd stress, maize growth was improved by colonization with AMF”. Data shows that Cd exposure at this concentration has only a negative effect on plant height. It is clear than inoculation has beneficial effect on maize when comparing to non-inoculated plants, but there is poor evidence of the AMF inoculation improving plant growth under 25 mg kg-1 Cd. Could you explain this discrepancy?
Lines 337 to 339. Authors claim that “Under 25 mg kg−1 Cd stress, AMF colonization had no significant effects on pectin content but significantly increased the lignin content in the cell walls of maize roots.” But looking at figure 3D, we can see that inoculation with the AMF increases pectin content when comparing with the control, Cd increases pectin content when comparing with the control and the inoculation with F. mosseae in the presence of Cd increases pectin content when comparing with the control. Could you explain this discrepancy between the text and data?

An English revision would be needed to improve the clarity of the text.
Round 2
Reviewer 2 Report
Review report ID jof-2504783_10072023_2:
Summary
Title: The role of lignin in the compartmentalization of cadmium in maize roots is enhanced by mycorrhiza
The paper describes how arbuscular mycorrhizal fungi (AMF) play a crucial role in the root systems of plants. The authors conducted a study to investigate how Cd is stored in the cell walls of maize roots colonized by Funneliformis mosseae. Their findings indicate that inoculating the roots with AMF lowers the amount of Cd in the maize shoots (63.6 ± 6.54 mg kg−1 vs. 45.3 ± 2.19 mg kg−1, p < 0.05) (28%) by retaining more Cd in the mycorrhized roots (224.0 ± 17.13 mg kg−1 vs. 289.5 ± 8.75 mg kg−1, p < 0.01) (29%). Additional research on the subcellular distribution of Cd showed that AMF colonization significantly improves the compartmentalization of 88.2% of Cd in the cell walls of maize roots, compared to the 80.8% of Cd associated with CWs in non-mycorrhizal controls. They claim that the presence of AMF did not increase the amount of Cd in pectin, a primary binding site for cell walls. However, it enhanced the content of lignin and the proportion of Cd in the total CWs-polysaccharides of the root cell walls. This finding was consistent with the increased activity of lignin-related enzymes, such as PAL, 4CL, and laccase which were also positively impacted by mycorrhizal colonization.
General concept comments
The Article writing has been improved significantly. The authors have addressed all minor comments and replied to mayor comments satisfactorily.
Minor comments
Lines 111, 258... Please replace Cd stress by “Cd exposure”. Please make sure this is corrected thoroughly in the manuscript.
In Table S2 line 11. Please replace “hemicellulose 2 (HC1)” by “HC2”.
